# The role of memory-dependent friction and solvent viscosity in isomerization kinetics in viscogenic media

Benjamin A. Dalton [1], Henrik Kiefer [1] & Roland R. Netz [1] ✉

Molecular isomerization kinetics in liquid solvent depends on a complex interplay between the solvent friction acting on the molecule, internal dissipation effects (also known as internal friction), the viscosity of the solvent, and the dihedral free energy profile. Due to the absence of accurate techniques to directly evaluate isomerization friction, it has not been possible to explore these relationships in full. By combining extensive molecular dynamics simulations with friction memory-kernel extraction techniques we consider a variety of small, isomerising molecules under a range of different viscogenic conditions and directly evaluate the viscosity dependence of the friction acting on a rotating dihedral. We reveal that the influence of different viscogenic media on isomerization kinetics can be dramatically different, even when measured at the same viscosity. This is due to the dynamic solute-solvent coupling, mediated by time-dependent friction memory kernels. We also show that deviations from the linear dependence of isomerization rates on solvent viscosity, which are often simply attributed to internal friction effects, are due to the simultaneous violation of two fundamental relationships: the Stokes-Einstein relation and the overdamped Kramers prediction for the barrier-crossing rate, both of which require explicit knowledge of friction.

Molecular conformational transition rates, such as the photo-isomerization rates of small molecules or the folding rates of proteins, are influenced by interactions between the molecule and its solvent environment, as well as by intra-molecular interactions within the molecule itself. In experimental settings, the molecular conformation dynamics of a solute molecule can be modulated by altering the viscosity $\eta$ of the solvating medium. This can be achieved by varying either the temperature or pressure of a mono-component solvent, or by systematically changing the concentration of a viscogenic agent, such as glucose, sucrose, or glycerol, in mixed aqueous solvents. The former approach was employed in early experiments probing the viscosity dependence of the picosecond-time-scale relaxation dynamics of small isomerizable molecules, such as stilbenes[1–5] or aromatic dimers[6,7]. The latter approach has played a critical role in the field of protein folding in uncovering the importance of internal friction effects[8–15]. Applying similar methodologies, all-atom simulations have been used to elucidate the molecular mechanisms of internal friction[16–21]. Typically, internal friction effects are characterized by the viscosity dependence of some molecular reconfiguration time scale $\tau$, such as the folding time of a protein or an isomeric-state transition time. In general, one assumes that $\tau = \alpha \eta^\beta + \varepsilon$ and argues that $\beta = 1$ and $\varepsilon = 0$ in the absence of internal friction and that either $\beta < 1$ or $\varepsilon > 0$ when internal friction effects are present. The supposed linear relation between $\eta$ and $\tau$ (i.e. $\beta = 1$) with $\varepsilon = 0$ is actually founded on the combination of two fundamental relations, which have hitherto been impossible to check separately. According to the Stokes-Einstein relation, the friction $\gamma$ acting on a molecule that moves through a solvent of viscosity $\eta$ satisfies $\gamma \sim \eta$, with a pre-factor that incorporates information about the molecule's geometry and surface slip. For sufficiently overdamped systems, Kramers' theory[22] shows that the average time $\tau$ to undergo a state transition by overcoming an energy barrier satisfies $\tau \sim \gamma$, with an exponential factor that depends on the

[1]Freie Universität Berlin, Fachbereich Physik, Berlin, Germany. ✉e-mail: rnetz@physik.fu-berlin.de

energy barrier. Therefore, the linear relation between $\tau$ and $\eta$ is indirectly mediated by the friction acting on the reconfiguring molecule, with deviations from linearity indicating violations of either the Stokes-Einstein relation, the overdamped Kramers relation, or both. While it is relatively commonplace to measure transition times and solvent viscosities, both in experiments and in simulations, a direct evaluation of the friction acting on some collective reaction coordinate is far more complicated. Therefore, the direct framing of the breakdown of $\tau$ - $\eta$ in terms of the underlying violations of the Stokes-Einstein relation or the overdamped Kramers relation for molecular isomerization in complex viscogenic solvents has so far not been possible.

To evaluate friction, one has in the past typically relied on indirect methods using memoryless reaction-rate theory[23–26]. In this article, we utilize recent non-Markovian memory kernel extraction methods[21,27–32] to directly evaluate the time-dependent friction acting on individual dihedrals in a variety of small isomerizing molecules. Memory extraction methods enable a direct evaluation of the friction acting on any arbitrary reaction coordinate by mapping the time series evolution of that reaction coordinate onto a generalized Langevin equation (GLE)[33,34]. These methods are remarkably general and have been applied recently to isomerization dynamics[21,35], cell migration[28], the vibrational spectra of water molecules[30], pair reactions in water[31], the dynamics of small polypeptide chains[29], and, most recently, to the folding dynamics of a diverse set of fast-folding proteins[32,36]. By directly extracting the friction, we open up the opportunity to parametrize various reaction-kinetic models and hence predict the corresponding reaction kinetics in comparison to simulation results. We simulate four n-alkane chains: n-butane, n-hexane, n-octane, and n-decane (hereafter, we omit the n prefix), and two capped amino acid residues, alanine and phenylalanine, both with NMA C-terminal capping and ACE at N-terminal capping, using molecular dynamics (MD) simulations with explicit solvents. In general, we consider the angle of specific dihedrals as a reaction coordinate. Previous work has shown that the dihedral angle itself may not provide an ideal reaction coordinate for describing the isomerization process of small photo-isomerizable molecules and that some more intricate reaction coordinate involving either the motion of the molecular axis[5] or internal bond stretching[37] may be more suitable. Therefore, for the case of butane, in addition to tracking just the dihedral angle, we consider an additional reaction coordinate and hence analyze the dynamics of the intra-molecular separation between the two $CH_3$ groups for the butane molecule, which explicitly captures internal molecular vibrations that are orthogonal to the dihedral angle. As a viscogenic agent, we mix water and glycerol, varying the glycerol concentration to change the solvent viscosity. We compare our results to an idealized system where we modify the viscosity of a pure water solvent by scaling the mass of the water molecules[17,38–41]. By analyzing the isomerization dynamics of selected dihedrals from within the various solute structures in the context of these solvents, we systematically explore the role of solvent composition and solute size in determining the relationships between dihedral friction, solvent viscosity, and isomerization kinetics.

Overall, we show a general violation of the Stokes-Einstein relation for dihedral isomerization and that the specific deviations from $\gamma$ - $\eta$ depend on the choice of the reaction coordinate. The isomerization times exhibit dramatically different viscosity scaling depending on whether we vary the glycerol concentration or the water mass, indicating the importance of solvent type and composition. This result is independent of reaction coordinate and is most extreme for the smallest solute butane, reducing with increasing solute size and hence systematically revealing solute-size effects. When evaluating the dependence of $\tau$ on $\gamma$, we find that $\tau$ - $\gamma$ does not hold and that $\tau$ is significantly reduced compared to the Kramers prediction in the overdamped limit. This dramatic acceleration of reconfiguration kinetics was recently reported for an extensive set of fast-folding

protein simulations[32], where it was shown that many proteins fold and unfold in a memory-induced barrier-crossing speed-up regime[42,43]. We conclude that the same non-Markovian mechanism also applies to dihedral isomerization kinetics and hence reveal the full complexity of the coupling between non-Markovian friction, solvent viscosity, and molecular reconfiguration kinetics in complex viscogenic media.

## Results

### Viscosity dependence of butane isomerization kinetics

We modify solvent viscosity by either varying the concentration of glycerol in a mixed water-glycerol solvent (denoted as w/gly) or by scaling the mass of the water molecules in a pure water solvent (referred to as super-heavy water and denoted as $\Delta m_w$). In the case of the super-heavy water, we uniformly change the mass of the water molecules such that the viscosity $\eta$ scales as $\eta/\eta_0 = \sqrt{m/m_0}$, where $m_0$ and $\eta_0$ are the mass and viscosity of neat water and $m$ is the scaled water mass[38–41]. This is an idealized approach with no experimental counterpart that is often used to simulate both high and low-viscosity solvents and has been essential for simulation studies investigating internal friction effects[17,19,21]. It is a key finding of the current article that care must be taken when interpreting kinetic results generated using the super-heavy water method. In Fig. 1a, we show a snapshot from MD simulations of a single united-atom butane molecule suspended in a water-glycerol solvent environment (see Methods and Supplementary Note 1 for simulation details). Butane is the smallest molecule to exhibit distinct isomeric states and is a classic model in studies of dihedral barrier-crossing processes[44–48]. The isomeric configuration of the butane molecule stochastically transitions between the trans- and gauche-states, driven by collision interactions with the water and glycerol molecules, which we register by tracking the dihedral angle $\theta$ and the intra-molecular distance $d_{14}$ (Fig. 1b). In Fig. 1c, we show typical 0.5 ns trajectory segments of $\theta(t)$ and $d_{14}(t)$ for butane in a neat water solvent. Isomeric state transitions occur by overcoming barriers in the free energy landscape. We extract free energy profiles from the trajectory of, for example, $\theta(t)$, such that $U(\theta) = -k_BT \log[\rho(\theta)]$, where $\rho(\theta)$ is the probability density and $k_BT$ is the thermal energy. $d_{14}$ is treated in the same way. In Fig. 1d, we show three example free energy profiles extracted from simulations for both $\theta$ and $d_{14}$: butane in standard pure neat water, in a super-heavy water solvent with $m/m_0 = 100$, and in a water-glycerol solvent with 60% mass-fraction glycerol. The three free energy profiles overlap, indicating that for the model employed in this study neither viscogenic method significantly affects the equilibrium properties of the dihedral. By symmetry, the two gauche configurations appear as a single state as a function of $d_{14}$. For the mass-scaled systems, we consider $m/m_0 = 1$, 9, 25, and 100, corresponding to $\eta/\eta_0 = 1$, 3, 5, and 10. Standard pure water for the TIP4P/2005 water model has viscosity $\eta_0 = 0.86$ mPas[49]. For the water-glycerol mixtures, we evaluate the viscosity using the Green-Kubo relationship, which relates the shear viscosity to auto-correlations of the shear stress tensor (Supplementary Note 2). In Fig. 1e, we plot the water-glycerol viscosities for the range of glycerol concentrations used throughout this paper and find excellent agreement compared to an experimental empirical curve for water-glycerol mixtures at $T = 300$ K[50].

To evaluate the friction, we map the projected reaction-coordinate trajectories onto a generalized Langevin equation (GLE)[33,34]:

$$m\ddot{q}(t) = -\int_0^t \Gamma(t - t')\dot{q}(t')dt' - \nabla_q U(q(t)) + F_R(t), \qquad (1)$$

where $q(t)$ is either $\theta(t)$ or $d_{14}(t)$. $\Gamma(t)$ is the friction memory kernel and $F_R(t)$ is the stochastic force term, which has a zero mean $\langle F_R(t)\rangle = 0$ and satisfies the fluctuation-dissipation theorem $\langle F_R(t)F_R(t')\rangle = k_BT\Gamma(t - t')$.

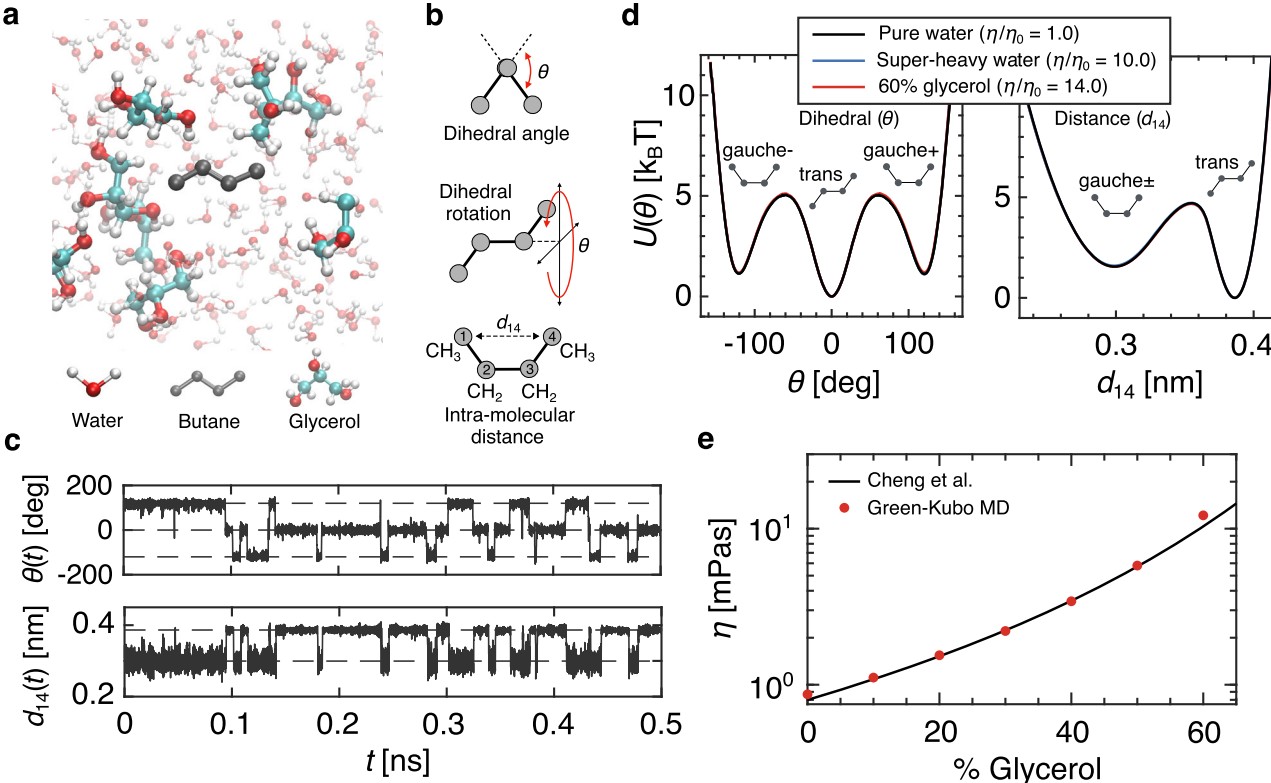

**Fig. 1 | Simulation of butane in explicit solvents. a** Simulation snapshot of a butane molecule dissolved in a water-glycerol mixture at 20% mass-fraction glycerol. **b** Schematic indicating the butane dihedral angle $\theta$ and the intra-molecular separation distance $d_{14}$ between the first and fourth carbon-hydrogen group. **c** A 0.5 ns trajectory segment for $\theta(t)$ and $d_{14}(t)$ in a standard pure water solvent. The dashed lines indicate the locations of the corresponding free energy minima.

**d** Free energy profiles for $\theta$ and $d_{14}$ extracted from MD simulations for three different solvent conditions. $\eta$ is the solvent viscosity. The schematics indicate the configurations of the gauche ± states and the trans state, which occupy the various energy minima ($T$ = 300 K). **e** Viscosity for water-glycerol mixtures, calculated using the Green-Kubo method, plotted as a function of glycerol mass fraction. Results are compared to an experimental empirical curve by Cheng et al.[50].

$U(q)$ is the corresponding free energy profile, as given in Fig. 1d, and $\nabla_q \equiv \partial/\partial q$. $m$ is the effective reaction-coordinate mass. It is known that a slight $q$-dependence of $m$ has little effect on the dynamics[51] (see Supplementary Note 3), therefore we assume $q$-independent mass.

We use recent friction memory-kernel extraction techniques and extract the running integral function $G(t) = \int_0^t \Gamma(t')dt'$ directly from the time series of $\theta(t)$ and $d_{14}(t)$[27,29]. Details of the memory kernel extraction method are given in Supplementary Note 4. In Fig. 2a, c, we show $G(t)$ under three solvent conditions. The fitting results for each curve, underlaid in grey, are discussed below. In Fig. 2b and d, we show the corresponding memory kernels $\Gamma(t)$, evaluated using numerical differentiation of $G(t)$. The large oscillations in $\Gamma(t)$ are due to bond angle vibrations, the latter being flexible in our model (Supplementary Note 5). The total friction $\gamma = G(t \to \infty)$ is given by the plateau values in Fig. 2a and c. In Fig. 2e, we show $\gamma$ for all solvent conditions as a function of the normalized viscosity $\eta/\eta_0$, where $\eta_0$ is the viscosity of neat water. For $\theta(t)$, we see that the viscosity dependence of the friction is similar, whether measured in super-heavy water or the water-glycerol mixtures, and exhibits strong sub-linear scaling over the range of viscosity considered here, which indicates a strong violation of the Stokes-Einstein relation. The viscosity dependence of the total friction acting on $d_{14}$ depends strongly on the solvent type. However, neither dependence satisfies a purely-linear relationship $\gamma \sim \eta$, with $\gamma \to 0$ as $\eta \to 0$. Therefore, the Stokes-Einstein relation is violated for both reaction coordinates and solvent types.

In Fig. 2f, we show the mean first-passage times $\tau_{MFP}$ plotted as a function of solvent viscosity, trans states are defined at $\theta = 0$ deg and $d_{14} = 0.299$ nm and gauche states at $\theta = \pm 120$ deg and $d_{14} = 0.386$ nm, such that $\tau_{MFP}$ is evaluated between these positions. We detail the

calculation of $\tau_{MFP}$ in Supplementary Note 6, where we discuss a method for eliminating state-recrossing effects[52]. In super-heavy water, $\tau_{MFP}$ increases significantly with $\eta$. However, in the water-glycerol mixtures, $\tau_{MFP}$ is completely independent of solvent viscosity. This is an interesting result because it shows that the kinetics of butane isomerization are fundamentally different in two different solvent environments when measured over the same range of macroscopic viscosity. One possibility for the independence of $\tau_{MFP}$ on $\eta$ in the water-glycerol solvent could be that the butane dihedral rotation experiences a local viscosity that is different from the macroscopic viscosity, where we define macroscopic viscosity to be the result of the Green-Kubo relation (Fig. 1e), which has been suggested to occur for solutes that are small compared to the size of the viscogenic co-solvent[53,54]. Such effects can be independently quantified via the translational diffusion coefficients for the butane center of mass $D^{tr}$. For pure Stokes-Einstein translational motion, $D^{tr} \sim \eta^{-1}$ and deviations from linearity would suggest that the butane molecule experiences an inhomogeneous solvent environment. In the inset of Fig. 2e, we show that linearity holds in both the super-heavy water (as was shown previously[19]) and the water-glycerol mixtures, with slight deviations in the 60% glycerol solvent, demonstrating that the Stokes-Einstein relation does hold for translational motion and that the solvation environment around butane in water-glycerol mixtures is homogeneous, corresponding to the bulk solvent conditions (Supplementary Note 7 for details). In regards to $\tau_{MFP}$, it is interesting to note that while the relationship between $\gamma$ and $\eta$ depends so strongly on the choice of reaction coordinate, the reaction times $\tau_{MFP}$ measured for $\theta$ and $d_{14}$ are in precise agreement. Overall, the results for $\tau_{MFP}$ indicate that, in the small molecule regime, molecular conformation reaction kinetics can completely decouple from the macroscopic viscosity of the

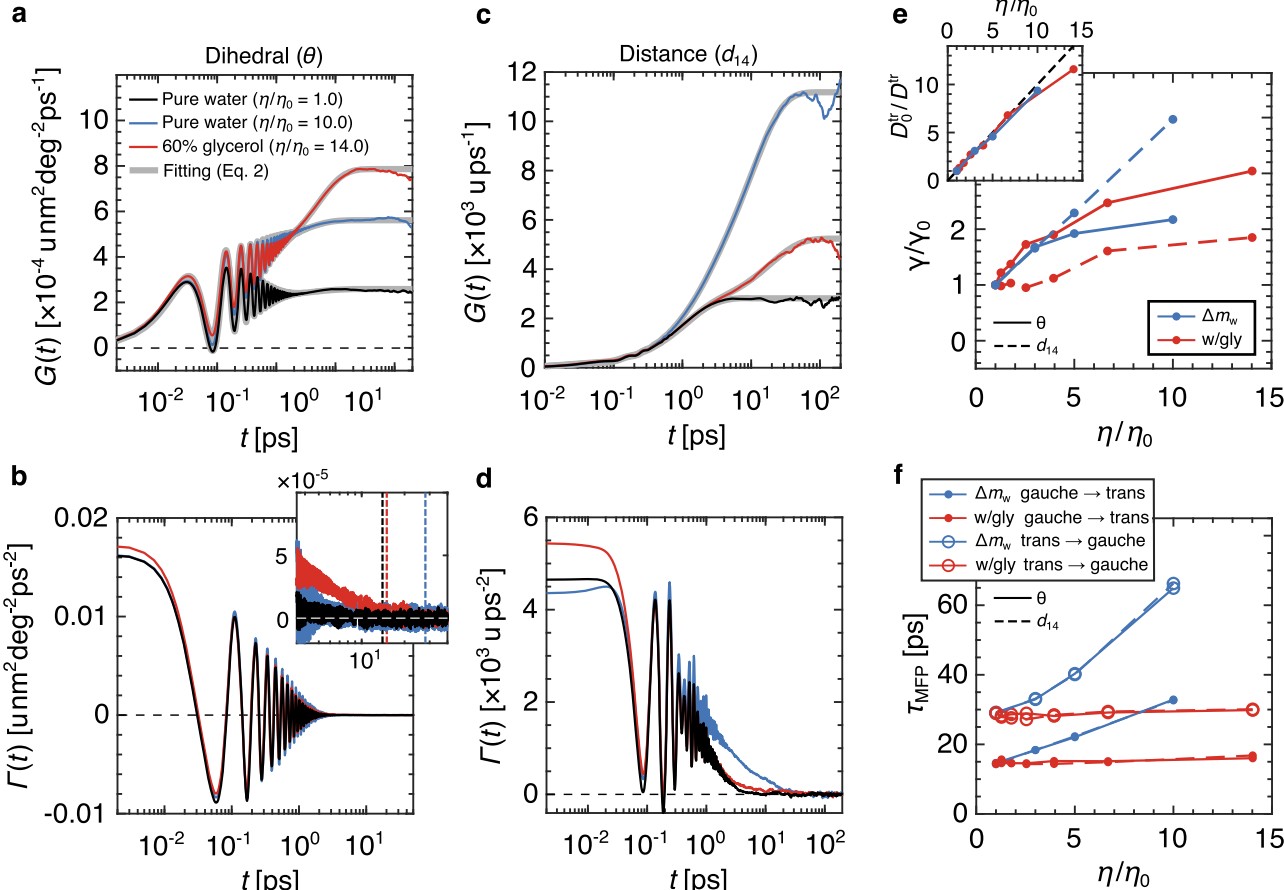

**Fig. 2 | Friction and isomerization kinetics for butane in viscogenic solvent environments. a, c** Running integrals $G(t)$ for the dihedral angle $\theta(t)$ and intramolecular separation $d_{14}(t)$, respectively, for butane isomerization under various solvent conditions, extracted from MD trajectories. The corresponding memory kernels $\Gamma(t)$ are shown in **b, d**, evaluated by numerically differentiating $G(t)$. Fits to $G(t)$ (Eq. (3)) are underlaid with the corresponding MD curves in **a, c**. The inset in **b** shows the long-time memory kernel decay, where the vertical dashed lines indicate the corresponding gauche → trans times. Figure legend in **a** also corresponds to **b–d**. **e** Total friction $\gamma$ acting on the butane isomerization for all superheavy water and water-glycerol conditions. The inset shows the inverse butane center of mass translation diffusion coefficient $D^{tr}$, where $D_0^{tr}$ is the result in standard neat water. The dashed line indicates linear proportionality. **f** Mean firstpassage times $\tau_{MFP}$ for the gauche-to-trans and trans-to-gauche transitions of the butane isomerization reactions.

solvent environment, despite the translational diffusivity remaining linearly coupled. This decoupling was discussed previously[47] when comparing solutes in a fluid solvent to solutes in a rigid matrix environment, where isomerization rates were calculated using the reactive-flux formalism[44], which has been applied previously to non-Markovian systems[55]. However, we show that the degree of decoupling depends on the solvent composition.

**Molecular-size dependence of isomerization kinetics**

To show that the decoupling between $\tau_{MFP}$ and $\eta$ in the water-glycerol solvent is a small-molecule effect, we systematically increase the length of the alkane chain to include hexane, octane, and decane and evaluate the mean first-passage times for the inner-most dihedral of each chain (Fig. 3a, see Supplementary Note 8 for details). Additionally, we simulate NMA-ACE capped alanine and phenylalanine amino acids and evaluate $\tau_{MFP}$ for the $\phi$-dihedral. $\tau_{MFP}$ for the alkanes exhibits a clear solute-size dependence (Fig. 3b). When rescaled by $\tau_{MFP}^0$ (the result for neat water), the isomerization times for both octane and decane show convergent scaling between the super-heavy water and water-glycerol mixtures at low viscosities. The results for alanine and phenylalanine are consistent with the alkane results since the backbone lengths of the two amino acids are both seven heavy atoms long, between that of hexane and octane (Fig. 3c). De Sancho et al. show that the viscosity scaling of isomerization kinetics in the super-heavy water

solvent is essentially the same for a range of dipeptides and that there are only slight deviations for the alanine dipeptide in the super-heavy water solvent when compared to a mixed glucose-water solvent[19]. However, they only measure in the range of $1 < \eta/\eta_0 < 3.5$, where they interpret the difference as negligible, suggesting that the scaling is therefore identical in the two viscogens (water + glucose and super-heavy water). Our investigation reveals that deviations are present, increasing with increasing viscosity.

In Fig. 3d, we show the dihedral isomerization times in neat water $\tau_{MFP}^0$. In Supplementary Fig. 7, we show that the relative change of $\tau_{MFP}^0$ with increasing alkane-chain length is similar for both the gauche → trans and trans → gauche transitions, with slight differences accounted for by the slight changes in barrier heights. The -140° → -70° transition in phenylalanine $\tau_{MFP}^0$ is much greater than that of alanine. This increase is only partially due to the 20% increase in the phenylalanine barrier height (Fig. 3c inset and Supplementary Note 8) with the remaining contribution coming from the presence of the large benzyl side group. In Fig. 3c, we see that the viscosity scaling for the two dipeptides is similar, suggesting that the addition of the large benzyl side group does not significantly affect the viscosity scaling of the isomerization times, but rather just the pre-factor. De Sancho et al. also addressed the issue of size dependence by expanding the radius of the united-atom groups in their n-butane model, which led to increased mean isomerization times[19]. Our results demonstrate that varying the width

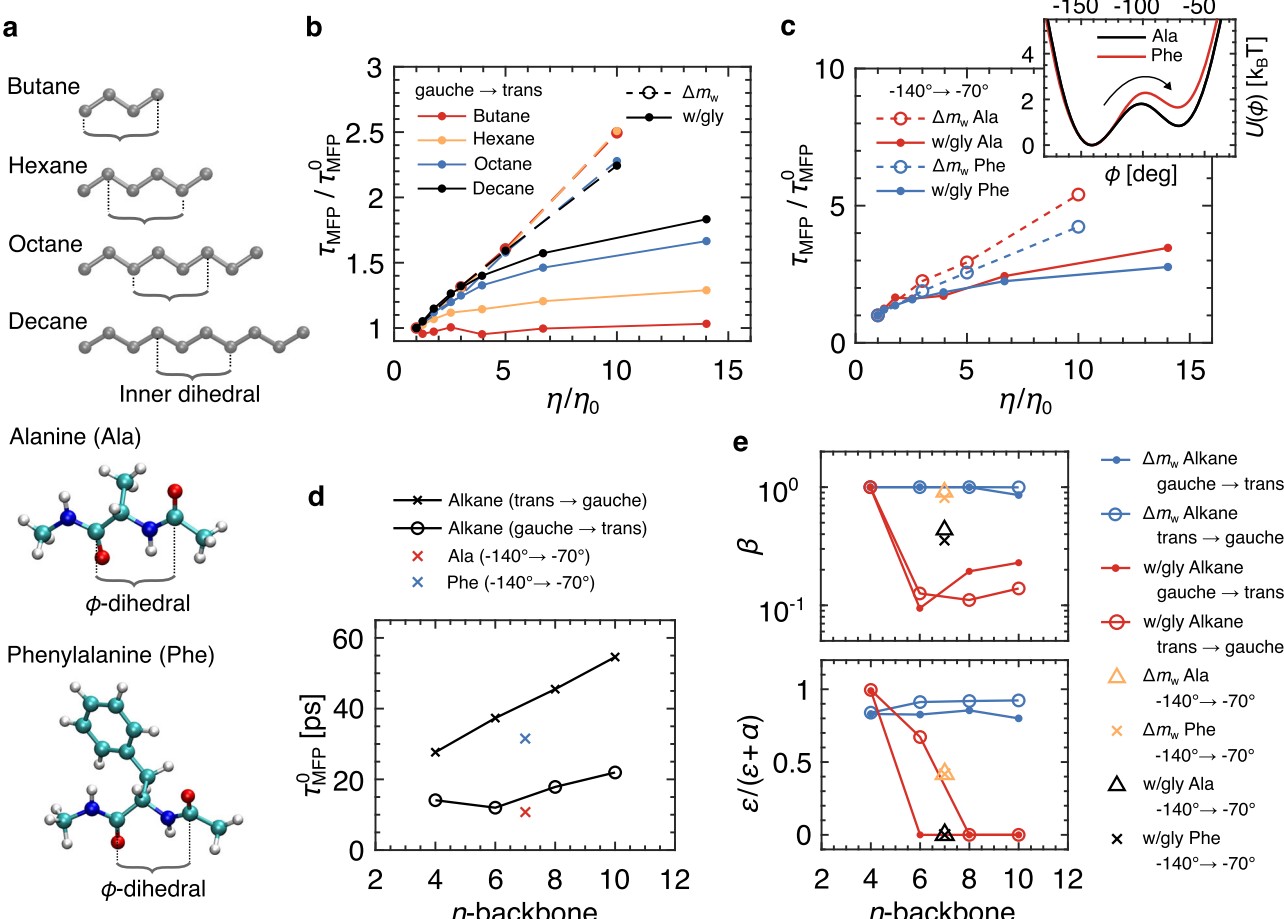

**Fig. 3 | Molecular-size dependence of isomerization kinetics. a** Molecular schematic for alkanes and two NMA-ACE capped amino acids. For the alkanes, the inner dihedrals are indicated. For alanine (Ala) and phenylalanine (Phe), the $\phi$-dihedrals are indicated. **b** Gauche-to-trans mean first-passage times $\tau_{MFP}$ for alkanes. $\tau_{MFP}$ values are scaled by the reaction times measured in neat water (pure water at standard-viscosity) for each system $\tau_{MFP}^0$. The left legend indicates the color scheme for the different alkanes. The right legend indicates the solvent type. **c** Scaled reaction times for NMA-ACE capped amino acids. Reaction times are shown for the $-140°$ to $-70°$ transition, as indicated by the arrow on the free energy profiles (inset, $T = 300$ K). **d** Reaction times in neat water $\tau_{MFP}^0$. Reaction times for alkanes are plotted as a function of backbone length (number of carbon atoms). For the capped amino acids, the backbone length is $n = 7$. **e** Fitting parameters $\beta$ and $\varepsilon/(\alpha + \varepsilon)$ for scaling relations: $\tau_{MFP}(\eta) = \alpha(\eta/\eta_0)^\beta + \varepsilon$ (See Supplementary Note 9).

of a molecule has a different effect compared to increasing a dihedral chain length, where it is the length of a dihedral backbone that predominantly influences isomerization rate scaling with viscosity.

We investigate the scaling relations shown in Fig. 3b, c in more detail by fitting $\tau_{MFP}$ for each molecule with the scaling function

$$\tau_{MFP}(\eta) = \alpha \left(\frac{\eta}{\eta_0}\right)^\beta + \varepsilon \qquad (2)$$

(see Supplementary Note 9 for details), which reveals two modes by which deviations from the simple Kramers-Stokes-Einstein scaling $\tau_{MFP} \sim \eta$ can manifest. The first is when $\varepsilon/(\alpha + \varepsilon) \to 1$, indicating that contributions from the non-zero intercept ($\varepsilon > 0$) dominate the prefactor $\alpha$. The second is when $\beta < 1$, such that the scaling of Eq. (2) is sublinear. In Fig. 3e, we show the exponent $\beta$ and the ratio $\varepsilon/(\alpha + \varepsilon)$ for all systems. For alkanes in super-heavy water (blue data), the scaling is effectively independent of chain length, with $\beta \approx 1$ indicating linear scaling for all systems. However, for all alkanes, $\varepsilon/(\alpha + \varepsilon)$ is between 0.8 and 0.9, indicating that despite the linear scaling, strong deviations from Kramers–Stokes–Einstein scaling, often interpreted in terms of internal friction effects, are present. For butane, it was previously shown that $\tau_{MFP}$ scales linearly with viscosity for $\eta > \eta_0$ but that the scaling transitions to an inertia-dominated regime for $\eta < \eta_0$[21]. We do

not consider this regime here. However, it is interesting to note that the approximate linear scaling for $\eta/\eta_0 > 1$ is a characteristic of all alkanes in super-heavy water. For alkanes in water-glycerol solvents (red data), $\varepsilon/(\alpha + \varepsilon)$ transitions from 1 to 0 for longer alkane chains, accompanied by a significant decrease in $\beta$. Here, butane is insensitive to changes in viscosity such that $\beta \to 1$ and $\alpha \to 0$ represent a constant function. However, chains longer than butane are sensitive to $\eta$ such that $\tau_{MFP}(\eta) = \alpha(\eta/\eta_0)^\beta$, with $\varepsilon = 0$ and sub-linear scaling such that $\beta < 0.25$. The dipeptides are interesting because they also scale linearly in the super-heavy water but with relatively reduced internal friction contributions ($\varepsilon/(\alpha + \varepsilon) \approx 0.4$). In contrast, in the water-glycerol solvents, the peptides are described by $\varepsilon = 0$ but with $\beta \approx 0.45$. Altogether, these results confirm two distinct behaviors for the viscosity dependence of dihedral isomerization kinetics and that it is not just the construction of the molecule or the viscosity of the solvent that determines the nature of the viscosity scaling, but also the composition of the solvent.

**Memory-induced speed-up of isomerization kinetics**

Since we directly extract $\gamma$ without fitting a particular kinetic model, we can compare different models for isomerization dynamics to the MD data. In Fig. 4a, we show the dependence of the gauche-to-trans $\tau_{MFP}$ for the butane and the inner decane dihedral on the extracted $\gamma$. For

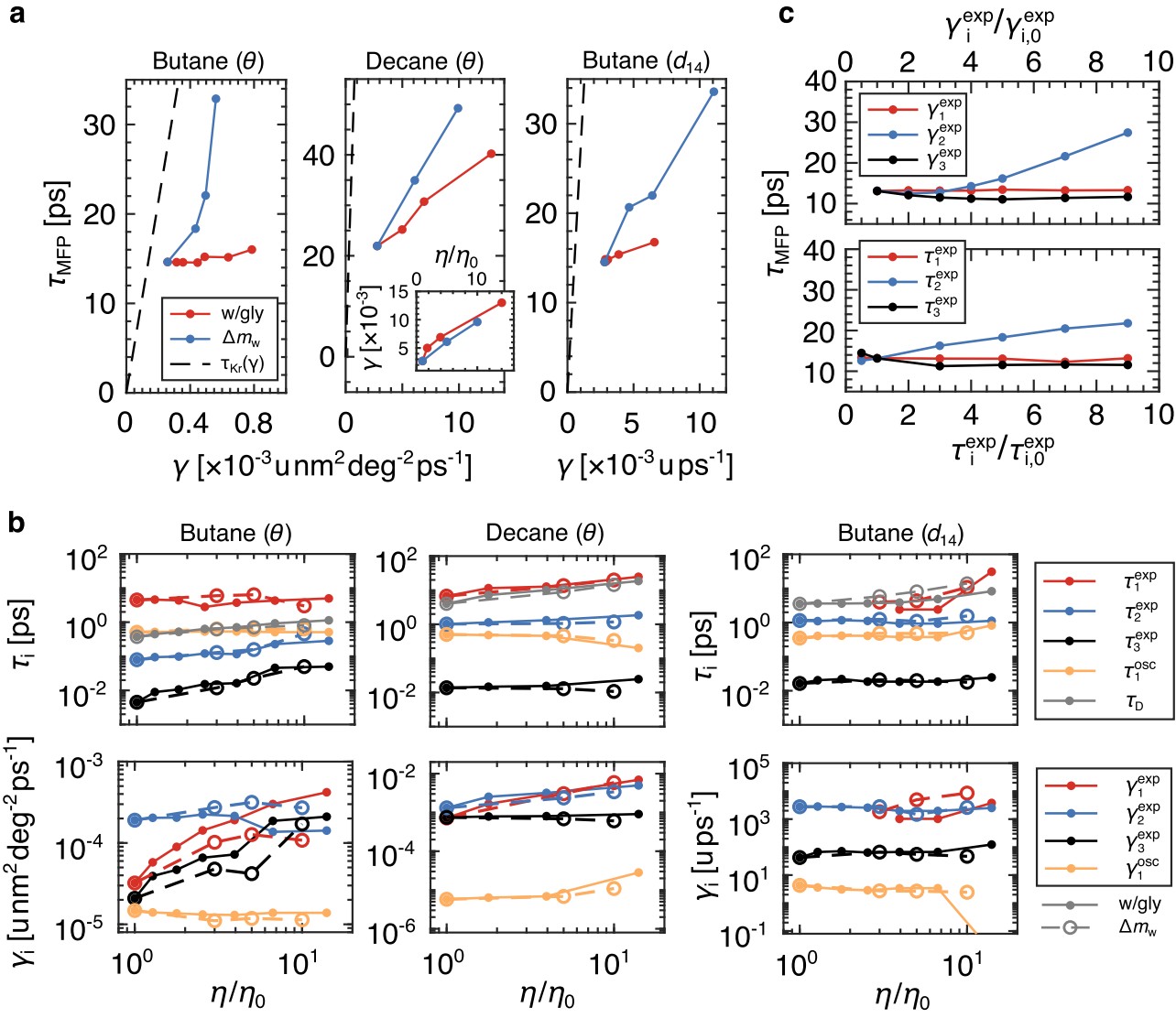

**Fig. 4 | Multi-time-scale friction and kinetic-model predictions of isomerization dynamics. a** Dependence of mean first-passage time $\tau_{MFP}$ on dihedral friction for butane and decane. $\tau_{Kr}(\gamma)$ is the overdamped, high friction Kramers prediction (black dashed line). The inset for decane shows the friction-viscosity scaling, which is shown for butane $\theta$ and $d_{14}$ in Fig. 2e (units are the same as the main panel). **b** Fitting parameters for Eq. (3) for fits of the memory kernels of butane and the inner dihedral of decane. $\gamma_i$ is for the set of normalized coefficients $\gamma_1^{exp}$, $\gamma_2^{exp}$, $\gamma_3^{exp}$, and $\gamma_1^{osc}$, written such that $\gamma_1^{exp} + \gamma_2^{exp} + \gamma_3^{exp} + \gamma_1^{osc} = \gamma$, where $\gamma$ is the total friction on the dihedral, and $\tau_i$ is for the set of corresponding time scales. The time scales are compared to the diffusion times $\tau_D = \gamma L^2 / k_B T$, where $L = 60$ deg. For $d_{14}$, an additional long-time-scale mode is excited for increasing viscosity, which we represent as $\tau_1^{exp}$. **c** $\tau_{MFP}$ for 1D GLE simulations with a three-component decaying exponential memory kernel, performed using Markovian embedding methods. The GLE model is parameterized for butane in neat water, with values taken from **b**, and the amplitudes and time scales are systematically perturbed around these values.

decane, we also show the friction-viscosity scaling (inset). In the high friction, overdamped limit, Kramers' theory predicts that $\tau_{Kr}(\gamma) = 2\pi\gamma e^{U_0/k_B T}/\sqrt{|U''_{min} U''_{max}|}$. For the butane dihedral angle, $\tau_{Kr}(\gamma)$ clearly does not represent the water-glycerol system well (Fig. 4a) and overestimates the super-heavy water times by as much as a factor of 3 (see Supplementary Table 2 in the Supplementary Note 10 for a table of parameters). For decane, the range of $\gamma$ is an order of magnitude higher than butane but the simulated $\tau_{MFP}$s are roughly the same, such that the Kramers prediction $\tau_{Kr}(\gamma)$ dramatically overestimates the measured values by a factor of between 13 and 25. A similar deviation is also seen for the butane intra-molecular distance, emphasizing the reaction-coordinate dependence of the prediction. It was previously shown that for overdamped Markovian systems, the prediction for the mean barrier crossing time in a smooth double-well potential with a barrier height of ~4$k_B T$ based on Kramers' theory deviates from the exact solution which incorporates the explicit free energy profile by

about 20% (see Supplementary Fig. 4 in the SI of ref. [31] and [56]), which is much smaller than the deviation we see in Fig. 4a. Overall, these results suggest that memory-induced barrier crossing speed-up effects are present. However, in the Supplementary Note 10, we show that the Grote-Hynes theory[57], which explicitly accounts for frequency-dependent friction effects, does not consistently predict the MD reaction times for butane and decane isomerization in both the water-glycerol solvent and the super heavy water.

To quantify memory effects, we fit the following function to the extracted memory kernels[30]:

$$
\begin{aligned}
\Gamma(t) \approx &\sum_{i=1}^{3} \frac{\gamma_i^{exp}}{\tau_i^{exp}} e^{-t/\tau_i^{exp}} \\
&+ \frac{(1 + \omega_1 \tau_1^{osc})\gamma_1^{osc}}{2\tau_1^{osc}} e^{-t/\tau_1^{osc}} \left[\cos(\omega_1 t) + \frac{\sin(\omega_1 t)}{\omega_1 \tau^{osc}}\right].
\end{aligned}
\tag{3}
$$

Here, $\gamma_i^{\mathrm{exp}}$ and $\tau_i^{\mathrm{exp}}$ are the amplitudes and time scales for exponentially decaying modes. $\gamma_1^{\mathrm{osc}}$ is the amplitude, $\omega_1$ is the angular frequency, and $\tau_1^{\mathrm{osc}}$ is the decay time for the decaying-oscillating term. Eq. (3) is written such that $\gamma_1^{\mathrm{exp}} + \gamma_2^{\mathrm{exp}} + \gamma_3^{\mathrm{exp}} + \gamma_1^{\mathrm{osc}} = \gamma$. Example fits of Eq. (3) are shown in Fig. 2a and c, and Supplementary Figs. 15 - 17. It has been shown that for systems with multi-exponential memory kernels, memory components with times scales much longer than the diffusion time $\tau_D = \gamma L^2 / k_B T$ do not affect mean first-passage times[43,58]. Here, $L$ represents a characteristic length, taken to be either 60 deg or $5.6 \times 10^{-2}$ nm, the distance from the gauche minimum to the barrier top. For butane, the longest exponential time scale ($\tau_1^{\mathrm{exp}}$) is much greater than $\tau_D$ (Fig. 4b). Therefore, the faster exponential modes ($\tau_2^{\mathrm{exp}}$ and $\tau_3^{\mathrm{exp}}$) have the dominant influence on $\tau_{\mathrm{MFP}}$. The amplitudes $\gamma_2^{\mathrm{exp}}$ are greater in the super-heavy water than in the water-glycerol mixtures, which helps to explain the divergent behavior for $\tau_{\mathrm{MFP}}$ in Fig. 2d. We idealize this effect with 1D GLE simulations over the dihedral free energy profile given in Fig. 1d, with a multi-exponential memory kernel parametrized using the extracted values in Fig. 4b (see Kappler et al.[43] for details of Markov-embedding simulations). Taking the values for neat water, we systematically perturb the individual $\gamma_i^{\mathrm{exp}}$ and $\tau_i^{\mathrm{exp}}$ values and show that the reaction times are only sensitive to changes in $\gamma_2^{\mathrm{exp}}$ and $\tau_2^{\mathrm{exp}}$ for this range of parameters (Fig. 4c). While the range of $\gamma_2^{\mathrm{exp}}$ required to modulate $\tau_{\mathrm{MFP}}$ is greater than the range traversed in Fig. 4b, it is essential to note the selectivity of $\tau_{\mathrm{MFP}}$ to specific memory components. In regard to the oscillating mode in Fig. 4b, $\tau_1^{\mathrm{osc}}$ is approximately equal to $\tau_D$ but the amplitudes $\gamma_1^{\mathrm{osc}}$ are relatively small. In Supplementary Note 12, we investigate the influence of the oscillating contributions to the Grote-Hynes prediction and show no effect on the viscosity scaling of the barrier crossing times. For decane, the longest exponential mode is likely the dominant contribution since $\tau_1^{\mathrm{exp}} \approx \tau_D$. Interestingly, the increase in viscosity excites a long time scale component in the $d_{14}$ dynamics in butane (identified as $\gamma_1^{\mathrm{exp}}$), which is similar to $\tau_D$, and which has a greater amplitude in the super-heavy water.

Overall, this time-scale analysis shows that dihedral dynamics are governed by friction memory effects and that isomerization kinetics occupy a memory-induced speed-up regime. When considering dihedral angles, decane exhibits strongly accelerated kinetics (Fig. 4a), which is expected for systems with $\tau_1^{\mathrm{exp}} \approx \tau_D$[43,58] since such systems are most sensitive to kinetic speed-up effects. As memory time scales exceed $\tau_D$, systems approach a memory-induced slow-down regime, which is the kinetic mode for some proteins[32]. For the butane dihedral angle, speed-up effects are present but weaker, suggesting that butane is closer to the memory-induced slow-down transition, which is consistent with the measurement that $\tau_1^{\mathrm{exp}} > \tau_D$. This interpretation is consistent for butane $d_{14}$. Overall, these results reveal that dihedral isomerization in viscous solvents exhibits multi-time-scale non-Markovian dynamics with memory-accelerated isomerization kinetics.

## Discussion

We have shown that deviations from the linear relation between isomerization times and solvent viscosity $\tau \sim \eta$ are simultaneously ascribed to violations of the Stokes-Einstein relation for dihedral rotation in solvent and Kramer's theory for the barrier crossing kinetics of an overdamped reaction coordinate. We arrive at this conclusion by applying memory-kernel extraction techniques, based on the generalized Langevin equation (Eq. (1)), to two reaction coordinates that track the trans ↔ gauche dynamics of an isomerizing n-alkane dihedral, and hence directly extract the corresponding frictions. Overall, we have systematically shown that solute size, solvent composition, and the choice of reaction coordinate all contribute to the breakdown of $\tau \sim \eta$, and that friction memory effects are a determining feature of dihedral isomerization dynamics, thereby revealing the full complexity of solvent-to-solute-reconfiguration coupling.

The viscosity-dependent isomerization times of larger molecules, such as decane or NMA-ACE capped amino acids, appear to converge between the two solvation methods, at least in the lower viscosity regime (Fig. 3b, c). The dependence on solvent type, rather than just viscosity, has been reported for larger molecules. For example, evidence suggests that the viscosity scaling of relaxation rates remains largely consistent for hairpin-forming polypeptide chains, regardless of whether they are solvated in a glucose or sucrose co-solvent. However, deviations have been observed for helix-forming polypeptide chains[9]. Similarly, Sekhar et al. showed that the interconversion rates of a four-helix bundle domain are different when measured in either a mixed water-glycerol solvent or a mixed water-bovine serum albumin (BSA) solvent[54], which they attributed to micro-viscosity effects. Typically, one distinguishes between macro-, micro-, and nano-viscosity. Macro-viscosity refers to the viscosity measured by a rheometer or, in our case, the Green-Kubo relation (Supplementary Note 2). Micro-viscosity refers to the viscosity experienced by micrometer-sized colloidal particles, measured, for example, by dynamic light scattering[59]. Nano-viscosity, then, refers to the viscosity experienced by a single probe molecule as it moves in a molecular medium. The motion can be translational[60] or rotational[61–65]. We show that for the case of the smallest solute butane, the translational motion is well described by the Stokes-Einstein relation using the macroscopic solvent viscosity. This holds even in the water-glycerol solvents, where one might expect deviations due to inhomogeneous-solvent effects[66–68]. Rotational nano-viscosity effects are directly relevant in the study of molecular rotor dyes, where the reconfiguration kinetics of a dye molecule are used to estimate the local viscosity at the nano-scale[69–73]. These fluorescent molecules undergo stochastic isomeric switching at rates determined by the local viscosity at the nano-scale. As we show here, non-Markovian multi-time-scale friction effects are important for understanding the rotational dynamics of these dyes.

Finally, we mention that the choice of collective variable used to represent a reaction coordinate plays a central role in the study of activated barrier crossing and the impact of memory and friction effects will be assessed differently depending on this choice. Methods exist for identifying optimal reaction coordinates[74–77]. However, it is unclear whether such methods simultaneously minimize non-Markovian effects[32,78]. It is known that when solvent relaxation time scales are slow compared to reaction times, then collective solvent coordinates must also be considered if a Markovian description is desired. This was originally introduced in Marcus's theory of electron transfer[79] and is well known in relation to aqueous proton transfer[80]. In Supplementary Note 13, we show that solvation relaxation times around the butane molecule are of the same order as, or slightly longer than, the longest dissipative time scales detected by the pure solute coordinates $\theta$ and $d_{14}$. Methods also exist for treating multi-dimensional reaction coordinates[81–83]. However, by analyzing arbitrary one-dimensional reaction coordinates with accurate memory-kernel-extraction methods, as we have done in the present article, one intrinsically captures the dynamics of the orthogonal degrees of freedom in the friction memory kernel. This is particularly relevant in the context of experiments, for example, where one does not have the luxury of choosing optimal or advanced reaction coordinates. Therefore, our analysis of intuitive, pure-solute coordinates is of fundamental importance.

## Methods

We carried out all simulations using the GROMACS simulation package (version 2020) and employed the TIP4P/2005 rigid water model[84]. To model glycerol and alkane molecules, we used the GROMOS53A6 force field[85] and represented all n-alkanes as united-atom chains. We neglected non-bonded 1-4 Lennard-Jones interactions and compensate by including a Ryckaert-Bellemen dihedral potential[86]. For glycerol

molecules, we explicitly represented all atoms, including hydrogens. We constrained all bond lengths using the LINCS constraint algorithm[87]. We allowed all bond angles to vibrate, including the hydrogen atoms in glycerol, except for the case of water. For the amino acid simulations, we studied single alanine and single phenylalanine molecules, each with NMA C-terminal capping and ACE at N-terminal capping, resulting in two peptide bonds for each molecule. We used the Amber99 force field, with all bond lengths fixed and all bond angles flexible. To ensure accurate simulation results, we pre-equilibrate all systems in the NPT ensemble with a Berendsen barostat[88] set to 1 atm. For production runs, we perform all simulations in the NVT ensemble with 2 fs simulation time steps and a temperature of 300 K, controlled with a velocity rescaling thermostat. For butane, we run individual simulations for 10 μs, which we require for convergence of the long-time-scale memory effects. We analyze data at full-time resolution. For simulations of longer alkanes (hexane, octane, and decane), we run individual simulations for 4 μs and analyze results at a resolution of 20 fs (decane inner dihedral is analyzed at full-time resolution for memory kernel extraction). For the amino acid simulations, we run individual simulations for 3 μs and analyze results at a full time-resolution. For the calculation of solvent viscosities, we run production simulations for 500 ns. For additional details regarding the analysis of simulations, see Supplementary Note 1.

## Data availability

Trajectory data for the butane dihedral angles and intra-molecular distance, as well as the decane dihedral angle, under all viscogenic conditions, have been deposited in the Zenodo database under accession code https://doi.org/10.5281/zenodo.10885500. All other data are available upon request.

## Code availability

All primary data was generated using GROMACS simulation package (version 2020). All input files and custom data analysis software are available upon request.

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

## Acknowledgements

The project was supported by the European Research Council (ERC) Advanced Grant 835117 NoMaMemo and the Deutsche Forschungsgemeinschaft (DFG) Grant No (R.R.N). SFB 1449 ID 431232613 "Dynamic Hydrogels at Biointerfaces" Project A02 (R.R.N). We would like to acknowledge the HPC Service of ZEDAT, Freie Universität Berlin, for providing computing time. We are also thankful to the physics department HPC services at Freie University of Berlin for their generous support.

## Author contributions

B.A.D. and R.R.N. designed research; B.A.D. performed research; B.A.D. and H.K. analyzed data; and B.A.D. and R.R.N. wrote the paper.

## Funding

## Competing interests

The authors declare no competing interests.
