## [Peer Review File · Nature Communications]

The role of memory-dependent friction and solvent viscosity in isomerization kinetics in viscogenic mediaReviewer #1 (Remarks to the Author):

The idea that we should see Stokes-Einstein behaviour at the level of torsional angle transitions of alkanes and peptides in water and water-glycerol mixtures is a bit of a straw man. While the authors note the work of Eaton and others on the relationship between bulk viscosity and friction on a reaction coordinate, there was substantial pre-dating that research by Troe, Eisinger, Hochstrasser, and others. For the systems studied, the observation of the breakdown of Stokes-Einstein relation, for relating solvent viscosity and friction experienced by a reaction coordinate, is not surprising. That said, how bulk viscosity and internal friction contribute to the friction acting on a reaction coordinate is an interesting topic for current research.

This work is ambitious and addresses important and long-standing unanswered questions related to the relative importance of internal friction and solvent friction on barrier crossing rates. The scholarship could be improved by recognizing a substantial body of work that addressed these questions in the past. I have a few comments below that I would ask the authors to consider in any revised version of this work. I would be happy to review a revised version of this work.

Nomenclature and potential of mean force

(1) In Figure 1, the authors refer to cis and trans states for butane. ButEne has cis and trans states, but butAne has trans and gauche+ and gauche- states. This should be corrected in the figure and discussion throughout the manuscript.

(2) I am confused by Figure 1(D). The caption says free energy profile labeled $U(\theta)$. The gas phase potential energy should be shown for comparison. Moreover, I see no dependence on the solvent in the profiles, although there is certainly dependence on solvent as was shown in many earlier studies.

Recognition of prior work on slowly reorganizing solvent

(3) The authors comment that "in the small molecule regime, molecular conformation reaction kinetics can completely decouple from the macroscopic viscosity of the solvent environment." This is well known. Please refer to work by Berne and coworkers on butane isomerization in CCl_4 .

(4) In the case of the glycerol-water solvent, the solvent relaxation time scales are slowed and the isomerizing solute undergoes reaction dynamics in the cage presented by the glassy solvent. This is not simply a mass effect. As such, the observation that the isomerization kinetics differs in glycerol-water and super-heavy water is unsurprising.

Impact of mass scaling on memory function(5) Relevant to the use of the super-heavy water model, did the authors observe whether the memory function displays a simple time scaling as would be predicted for a non-linear reaction coordinate in a bi-linearly coupled harmonic bath? That is the model employed by Zwanzig in his 1973 J. Stat. Phys. paper, the only example we have of the derivation of a generalized Langevin equation from a Hamiltonian system.

(6) Following the point made in (5), did the authors consider whether their reaction kinetics could be captured by a multi-dimensional transition state theory a la Pollak?

Choice of reaction coordinate

(7) I feel that the memory functions have a strange form. I believe this results from (1) the collective variable chosen as a reaction coordinate, which is simply the bare dihedral angle, and complications from the internal vibrational dynamics. The torsion angle tracks with the transition between trans and gauche in butane, but it is not a good reaction coordinate in the sense of transition state theory.

(8) The choice of reaction coordinate will have a strong impact on the assessment of the relative importance of internal friction. This should be discussed.

(9) The authors might consider a more careful identification of reaction coordinates for these transitions. For example, one might consider the recent work of Dinner and Ma, and the more recent work of Ma, for the identification of "exact" reaction coordinates.

(10) It is known that the angles that form the dihedral, which can significantly alter the barrier to transition between states, is an important element of the reaction dynamics and is missing from this model. Berne showed this in butane. Similar observations have been made for conformational transitions in proteins involving changes in torsional angles. See the early work of Wutrich (experimental) and Karplus (theoretical) on the tyrosine ring-flip in BPTI.

Choice of system-bath decomposition

(11) The choice of reaction coordinate as the "slow variable" will determine the partitioning of system and bath dynamics and the nature of the memory function. In cases where the solvent relaxation is slower than the time scale of the barrier crossing, the reaction coordinates may need to explicitly include solvent degrees of freedom. There are many examples of this, from Marcus theory to Agmon-Hopfield models of protein ligand dynamics to the proton transfer theory of Hynes that includes a solvent polarization coordinate. The authors should consider this alternative perspective which is commonly employed in theories of chemical reaction dynamics.

(12) The authors note, "However, as we have shown here, these difference may be rather due to complex non-Markovian effects that result from the interactions between the protein domain and mixed solvent environments." This statement again fails to recognize that in many cases of chemical dynamics in complex solvents, the collective variable chosen for the reaction coordinate may be a combination of protein and solvent coordinates.

Modest changes in rate

(13) I will close by noting that in many cases the variation in the rate over the range of viscosity studied is modest. It is very hard to confirm any functional dependence without having at least an order of magnitude variation in rate. This has been a long-standing challenge in the search for the Kramers turnover, that very large changes in solvent viscosity or density often leads to modest changes in rate. I would again refer the authors to earlier experimental work by Troe, Jonas, Eisinger, and others.

Reviewer #2 (Remarks to the Author):

This manuscript studies the role of friction on isomerization kinetics for several systems using a generalized Langevin equation model and reveals the importance of memory effects and violation of the usual inverse proportionality between solvent viscosity and isomerization rate. The viscosity change is introduced by either altering the mass of water or changing the composition of the solvent.

In general, I find this to be an insightful study, but I have a few questions.

1. On p. 3, where the authors discuss mean first passage times, they should explain, precisely, the initial conditions and the final state to which the mean first passage times are calculated.

2. The distinction between the "nano viscosity effect" that the authors dismiss and decoupling of the molecular kinetics from macroscopic viscosity that they authors argue is true is thoroughly confusing: to me they sound like the same thing. Is there a quantifiable statement one could make to differentiate between the two effects?

3. Figure 2C is unconvincing - deducing scaling laws when viscosity changes by merely an order of magnitude is dubious. For example, in the inset the authors argue that they see direct proportionality to viscosity, but one could equally claim some other power law, as some curvature is observed. And while all plots tend to look like straight lines on the Log-log scale, the main Fig. 2C doesn't even quite look like a straight line!

At the very least, the authors should provide both linear plots and log-log plots for both cases (γ and D^{tr}) to convince the reader that their interpretation is correct.

Reviewer #3 (Remarks to the Author):

Dalton et al. study the impact of non-Markov effects on the dynamics of dihedral angle rotations. The work focuses on understanding the relationships between internal friction, viscosity, and the free energy profile of a reaction. The authors used molecular simulations of butane and longer alkanes along with two dipeptides (Ala2 and Phe2) in explicit water. The viscosity was varied either by changing the mass of water or by adding glycerol, a known viscosogen. Using the dihedral angle as the coordinate, the authors extract the memory friction kernel from the simulations using a previously established protocol. The resulting friction kernels are decaying functions (with oscillations due to couplings of dihedral and bond angle oscillations). The long-term friction coefficient shows clear deviations from the Stokes-Einstein viscosity dependence, regardless of the method used to change the viscosity. Surprisingly, the cis-trans mean first passage (MFP) times show rather different viscosity dependencies. While the MFP times increase with increasing water mass, the addition of glycerol had no effect on the cis-trans transition times for butane. The authors show that increasing the size of the molecule (from butane to decane) restores the viscosity dependence in the case of glycerol. However, the obvious explanation for the different behavior in heavy water and glycerol, i.e. a breakdown of continuum hydrodynamics due to the significant size of the solvent (glycerol/water) compared to the solute (butane to decane), could be ruled out since translational diffusion and dihedral angle friction behave identically at long times between the two methods of viscosity change. Since the friction kernel and the potential are known, the authors compute the MFP times from rate theories (Kramers, Grote-Hynes) and show significant deviations between rate theory and simulations for butane and decane, with the simulations being much faster than the predictions. The authors conclude that the time-dependent friction kernel as a whole determines the kinetics and causes faster relaxation times than expected.

The study is carefully done and addresses interesting questions in biomolecular and chemical kinetics. Overall, I like the paper. However, I should mention that previous work, in particular that of de Sancho and Best (2014) (ref. 12), which was also highlighted by the authors, has addressed the same questions and reached similar conclusions. The new findings in this study are twofold: first, co-solutes have a different effect on MFP times than water mass scaling, and second, Grote-Hynes theory is unable to properly predict the rates of dihedral angle switching dynamics. However, it should also be noted that the deviations of Grote-Hynes from the simulation results are substantially smaller than those of Kramers theory, essentially not exceeding a factor of three. Overall, the conclusion that non-Markovian effects are partly responsible for internal friction effects in dihedral angle dynamics is not entirely new. This point has also been discussed for butane by one of the authors in a previous publication (ref. 14, Daldrop et al. PNAS, 2014), although I got the impression that the explanation of internal friction effects in that study was slightly different. Therefore, as much as I like the present study, I am not fully convinced that the results warrant publication in Nature Communications. I have a few comments that the authors may want to consider.

1. This is a technical comment. I had serious problems understanding the handling of re-crossing (S7). I understand that barrier re-crossing effects need to be handled carefully when computing FP times. However, the authors seem to be concerned about re-crossing of states (in free energy minima) as suggested in their explanation in S7. The explanation of why this is a concern was unclear to me and should be better explained. In this context, the value of the delay δt and examples of FP time distributions for the other cases (All-to-First, Delayed All-to-First) should be

given.

2. Do the authors have an explanation for the different scaling of the MFP time with the number of backbone atoms for cis-trans and trans-cis transitions in alkanes? It seems that cis-trans transitions are less sensitive to the number of backbone atoms. Yet, the potentials for the individual alkanes are very similar.

3. I find the analysis of the timescales in the friction kernel very interesting (Fig. 4B). However, the discussion accompanying this figure is rather qualitative. It should be possible to omit some of the decay components in the kernel and then use Langevin simulations to generate trajectories and from those MFP times to see which decay component really dominates the viscosity dependence. This would be particularly interesting for butane.

4. Related to point 3, the friction kernel contains information about the solvent. Have the authors thought about directly studying the solvent dynamics around, let's say, butane compared to bulk solvent? This would be helpful to understand what actual molecular processes in the solvent cause the individual components in the friction kernel.

Response to Referee's comments: "Conformational isomerization dynamics in solvent violates both the Stokes-Einstein relation and Kramers' theory"

Reply to Reviewer #1

The idea that we should see Stokes-Einstein behavior at the level of torsional angle transitions of alkanes and peptides in water and water-glycerol mixtures is a bit of a straw man.

It is a straw man insofar as it is also a straw man to discuss deviations from linear scaling of the reaction times with viscosity, $\tau \sim \eta$, which are amply discussed in the literature. We have simply redirected the discussion in order to focus on the fundamental relations that are assumed when discussing deviations from linear $\tau \sim \eta$. We believe that this is a worthy addition to the literature.

(1) In Figure 1, the authors refer to cis and trans states for butane. ButEne has cis and trans states, but butAne has trans and gauche+ and gauche- states. This should be corrected in the figure and discussion throughout the manuscript.

We thank the reviewer for pointing this out. We have now changed all instances of the word "cis" to "gauche", both in the main manuscript and in the SI document.

(2) I am confused by Figure 1(D). The caption says free energy profile labeled U(theta). The gas phase potential energy should be shown for comparison. Moreover, I see no dependence on the solvent in the profiles, although there is certainly dependence on solvent as was shown in many earlier studies.

The gas phase potential energy is simply the Ryckaert-Belleman dihedral potential, introduced in section S1 for the Supplementary Information. The model that we use throughout our study has vibrating bond angles. This is the same in the model Rosenberg, Berne, Chandler (*Chem. Phys. Lett.* 1980), where the authors also implement Ryckaert-Belleman dihedral potential. However, in that article, there is no discussion made of solvation effects and hence no comparison to the gas phase profile. Solvation effects are discussed by Rebertus, Berne, and Chandler (*J. Chem. Phys.* 1979), where bond angles are rigidly fixed. In this context, the conformational equilibrium is certainly shown to depend on the solvent. Other later studies that look at the solvent effects on equilibrium dihedral distributions also employ rigid-angle models (see for example: Ashbaugh et al., *Biophysical Journal*, 1999 and Travis and Searles, *J. Chem. Phys.*, 1980, which we now cite in our Supplementary Information). We have now included an additional figure in Section S5 of the Supplementary Information where we show that the free energy profile of the butane dihedral is only slightly modified from the gas phase for the flexible bond-angle model, but that there are significant modifications for an equivalent rigid-angle model. In regards to the addition of glycerol, it is not known *a priori* to what extent glycerol concentration changes the conformation equilibrium of bare dihedral angles. We show that, for a model including vibrating bond angles, the effects of glycerol are small. Therefore, we have been able to discuss the contributions of friction on the isomerization rates in a controlled way.

(3) The authors comment that "in the small molecule regime, molecular conformation reaction kinetics can completely decouple from the macroscopic viscosity of the solvent environment." This is well known. Please refer to work by Berne and coworkers on butane isomerization in CCl4.

Our suggestion that "... molecular conformation reaction kinetics can completely decouple from the macroscopic viscosity of the solvent environment" is made in relationship to the water-glycerol solvent and we therefore suggest that this is a general phenomenon for all "natural" viscogenic solvents, i.e. multi-component solvents where no mass scaling is employed. We explicitly show that this interpretation is unsuitable for the super-heavy water, suggesting that the failure of the super-heavy water to reproduce this result is a shortcoming of the technique and that one must consider kinetic results generated using super-heavy water carefully. One must also consider the comparison to the work of Berne and coworkers (Rosenberg, Berne, and Chandler, *Chem. Phys. Lett.* 1980) carefully. Their "high viscosity" solvent is a rigidly-fixed matrix of spatially-frozen atoms, such that no energy or momentum can be transferred between the solute and solvent. In this regard, their rigid matrix is actually the infinite mass limit of our super-heavy solvent. Since the isomerization times scale linearly with viscosity in super-heavy water, we should expect infinitely slow isomerization times, which is in contrast to the result of Rosenberg et al. The reason that they observe finite isomerization transitions in a rigidly frozen solvent is that all simulations are initiated from a transition state. Therefore, there is no account of the time spent in a given state before traversing the barrier,

which is accounted for by the mean first-passage time methods used by us. The authors calculate the transition rate using the population decay/reactive flux method of Chandler. This method has not been validated for non-Markovian systems. In Fig.R1 of this response letter, we show that the standard Chandler method of identifying a long relaxation times-scale in the population decay correlation breaks down at moderate memory times, which is the regime for butane isomerization. We have now included a reference to Berne and coworkers (Rosenberg, Berne, and Chandler, *Chem. Phys. Lett.* 1980). However, we suggest that many important questions remain.

Fig.R1: In the reactive flux method of Chandler, the decay rate $k(t)$, calculated via the population correlations $C_{HH}(t)$, should reveal a distinct plateau regime, which is associated with the phenomenological reaction rate. Plotted in terms of $\tau(t) = k^{-1}(t)$, where $k(t) = -d(C_{HH}(t)/C_{HH}(0))/dt$, the plateau will appear as a region of constant τ . Here, using 1D GLE simulations performed with the Markovian-embedding technique (Kappler et al. JCP, 2018), we compare $\tau(t)$ to the reaction times evaluated via the mean first-passage times (see Supplementary Information section S7). We show that for moderate memory times ($\tau/\tau_D > 0.01$, where τ_D is the diffusion time scale), it is impossible to identify a distinct plateau region in $\tau(t) = k^{-1}(t)$. Therefore, the applicability of Chandler's reactive flux method involving population-operator decay to non-Markovian systems is unclear.

(4) In the case of the glycerol-water solvent, the solvent relaxation time scales are slowed and the isomerizing solute undergoes reaction dynamics in the cage presented by the glassy solvent. This is not simply a mass effect. As such, the observation that the isomerization kinetics differs in glycerol-water and super-heavy water is unsurprising.

The surprising feature is not so much that the isomerization kinetics differ between the glycerol-water and super-heavy water, but rather that they are different when discussed in terms of their independently determined friction influence on the reaction coordinate: the super-heavy water scaling approaches linearity for increasing viscosity but is completely independent in the glycerol-water. Our updated arguments concluding that butane does not experience a reduced viscosity compared to the bulk due to inhomogeneous solvent effects (see comment 2 by Reviewer 2, line 257 in the updated manuscript, and Supplementary Information section S6), in combination with our result that the friction experienced by the dihedral angle θ is the same in both water-glycerol and super-heavy water (Fig.2E), indicate that the suggestion "... the isomerizing solute undergoes reaction dynamics in the cage presented by the glassy solvent" is not conclusive. The reviewer's own reasoning (see previous item) is that it is known that the reaction dynamics decouple from viscosity. However, this statement was made in consideration of results generated in a rigid matrix environment, which, as we have said, is the infinite mass limit of the super-heavy water, where the latter mediates linear coupling. Therefore, the super-heavy water system scales linearly, which is in contrast to results generated using a rigid matrix and simulations initiated from a transition state, and the water-glycerol system exhibits complete independence, despite the diffusivity of the translational motion coupling linearly to macroscopic viscosity. Overall, we disagree that these results are unsurprising.

(5) Relevant to the use of the super-heavy water model, did the authors observe whether the memory function displays a simple time scaling as would be predicted for a non-linear reaction coordinate in a bi-linearly coupled harmonic bath? That is the model employed by Zwanzig in his 1973 J. Stat.

Phys. paper, the only example we have of the derivation of a generalized Langevin equation from a Hamiltonian system.

The time scales for the memory kernel of the butane dihedral angle are shown in detail in Fig.S16. There, we show that for the butane dihedral angle in the super-heavy water solvent, the memory times scale approximately linearly with solvent viscosity, except that the longest time scale deviates strongly for high viscosities.

(6) Following the point made in (5), did the authors consider whether their reaction kinetics could be captured by a multi-dimensional transition state theory a la Pollak?

We certainly considered a multi-dimension analysis. However, it is the premise of the present article to discuss projections onto a single-dimensional reaction coordinate, such that all orthogonal dynamics are captured in the one-dimensional friction memory kernel. That said, we agree that this is a very interesting direction for future research. We now mention this in our extended discussion section with references to Pollak and recent work by Acharya and Bagchi on multi-dimensional Markovian diffusion.

(7) I feel that the memory functions have a strange form. I believe this results from (1) the collective variable chosen as a reaction coordinate, which is simply the bare dihedral angle, and complications from the internal vibrational dynamics. The torsion angle tracks with the transition between trans and gauche in butane, but it is not a good reaction coordinate in the sense of transition state theory.

We appreciate the reviewer's concerns over our single choice of reaction coordinate, i.e. the bare dihedral angle. The internal bond vibrations that we include in our model are orthogonal to the dihedral angle and therefore contribute to the internal friction and memory functions (see Supplementary Information S5). In the revised version, we have chosen to include the analysis of an additional reaction coordinate which explicitly includes contributions from the vibrating bond angle: the intra-molecular distance between the two CH₃ groups in the butane molecule (denoted as d_{14} throughout). This is an important and extensive addition to our work so we have reorganized many small sections of our revised article to include the analysis of d_{14} in parallel with our analysis of the dihedral angle of butane. The results highlight the subtle reaction-coordinate dependence of many results but also validate our principle arguments. We have extended our discussion section to include these points. We hope that the reviewer is satisfied with this addition in regard to their concerns over our choice of reaction coordinate.

(8) The choice of reaction coordinate will have a strong impact on the assessment of the relative importance of internal friction. This should be discussed.

This is now discussed. See our reply to point (7) above.

(9) The authors might consider a more careful identification of reaction coordinates for these transitions. For example, one might consider the recent work of Dinner and Ma, and the more recent work of Ma, for the identification of "exact" reaction coordinates.

The methods by Dinner and Ma would indeed be suitable for application to the dipeptide molecules explored briefly in our paper, which is the same system for which they introduce their methods. Furthermore, it is certainly an interesting subject to investigate, for example, the friction acting on the carefully identified reaction coordinates resulting from these methods and to consider the role of the solvent viscosity in this analysis. However, we believe that such an undertaking would overcomplicate our study and would deviate from the main purpose of the analysis in the relevant section of our manuscript. The main point of Fig. 3 in our paper, where we consider the torsion dynamics of the dipeptide molecules, is to systematically show that the viscosity scaling of the isomerization kinetics of individual dihedrals changes as a function of molecular size and that the complete decoupling of isomerization kinetics from macroscopic viscosity pertains only to the smallest solutes. In doing so, we consider a single observable that is consistent for all solutes. We note also that we do not consider the friction in this section of the paper and make no discussion of friction concerning dipeptide reconfiguration. Therefore, we believe that introducing either the automated genetic neural network methods of Dinner and Ma (J. Phys. B., 2005) or the generalized work functional method of Ma (J. Chem Theory Compute, 2022) to establish an optimal 1D reaction coordinate would overload our investigation. We emphasize that the investigation carried out in our manuscript does not require reaction coordinate optimization, but can be applied to any arbitrary reaction coordinate. We do agree, however, that

a discussion of the methods introduced in the papers of Dinner and Ma is essential for our review of the field. We have now referenced these works in the revised manuscript.

(10) It is known that the angles that form the dihedral, which can significantly alter the barrier to transition between states, is an important element of the reaction dynamics and is missing from this model. Berne showed this in butane. Similar observations have been made for conformational transitions in proteins involving changes in torsional angles. See the early work of Wutrich (experimental) and Karplus (theoretical) on the tyrosine ring-flip in BPTI.

We thank the reviewer for pointing out this subtle detail. It is certainly an important point that the bond angles which form the dihedral play an important role in the overall dihedral dynamics. Since we include bond vibrations in our MD simulations, they will contribute to the dynamics of the orthogonal degrees of freedom, and hence the friction kernel, for the bare dihedral reaction coordinate. We compare the kinetics and friction for MD models with and without vibrating bond angles in the Supplementary Information section S5. We now further clarify the importance of this point in that section.

(11) The choice of reaction coordinate as the “slow variable” will determine the partitioning of system and bath dynamics and the nature of the memory function. In cases where the solvent relaxation is slower than the time scale of the barrier crossing, the reaction coordinates may need to explicitly include solvent degrees of freedom. There are many examples of this, from Marcus theory to Agmon-Hopfield models of protein ligand dynamics to the proton transfer theory of Hynes that includes a solvent polarization coordinate. The authors should consider this alternative perspective which is commonly employed in theories of chemical reaction dynamics.

This is an important point. We mention that it is an essential feature of our analysis that solvent degrees of freedom are captured by the memory kernel that we extract from our simulations. We now explicitly mention the work of Chandler and coworkers on the role of solvent in ion-pair dissociation kinetics, as well as the pioneering work of Marcus regarding electron transfer.

(12) The authors note, "However, as we have shown here, these difference may be rather due to complex non-Markovian effects that result from the interactions between the protein domain and mixed solvent environments." This statement again fails to recognize that in many cases of chemical dynamics in complex solvents, the collective variable chosen for the reaction coordinate may be a combination of protein and solvent coordinates.

We thank the reviewer for clarifying this point. We have rewritten this specific section to avoid misconception and now discuss the role of including solvent degrees of freedom in the chosen reaction coordinate in the discussion section.

(13) I will close by noting that in many cases the variation in the rate over the range of viscosity studied is modest. It is very hard to confirm any functional dependence without having at least an order of magnitude variation in rate. This has been a long-standing challenge in the search for the Kramers turnover, that very large changes in solvent viscosity or density often leads to modest changes in rate. I would again refer the authors to earlier experimental work by Troe, Jonas, Eisinger, and others.

We agree with the review that any discussion of a functional dependence between rate and viscosity is questionable for such a narrow range of data. Here, as is often the case in experiments, we cannot control the range of rates since the rates are observables that depend on other control parameters. In our work, the fundamental control parameter is indeed the concentration of glycerol, which we cap at 60% for computational reasons. Therefore, as the reviewer points out, we are left to accept a modest change in rate. A key aspect of our work, however, is the extraction of memory kernels and the evaluation of friction for any arbitrary reaction coordinate. Since friction is not obtained by fitting a kinetic model, we can use it to parametrize a range of kinetic models and to hence make predictions. Despite the narrow range of rates measured in our simulations, we insist that it is still very important to compare simulation results to predictions based on Kramers theory and the Grote-Hynes theory for the extracted friction and free energy profiles, which, as we have discussed, are not in good agreement. We have now clarified this point in our discussion and refer to the experimental work indicated by the reviewer.

Reply to Reviewer #2

1. On p. 3, where the authors discuss mean first passage times, they should explain, precisely, the initial conditions and the final state to which the mean first passage times are calculated.

We have now explicitly stated this in the appropriate section of the manuscript.

2. The distinction between the "nano viscosity effect" that the authors dismiss and decoupling of the molecular kinetics from macroscopic viscosity that they authors argue is true is thoroughly confusing: to me they sound like the same thing. Is there a quantifiable statement one could make to differentiate between the two effects?

We thank the reviewer for pointing out the lack of clarity on this important point in our originally submitted manuscript. Let us first summarize the state of the art and what we tried to convey in the original submission and after that describe the revised explanation in the new version of our paper.

According to the operational literature definition, the term nano-viscosity refers to the viscosity experienced by single probe molecules as they move through a molecular medium. The motion can be translational (experimentally detected for example by fluorescence correlation spectroscopy [see reference **58** in the current submission]) or rotational (experimentally detected for example by fluorescence anisotropy decay, electron paramagnetic resonance [**59**], NMR relaxation [**60**], fluorescence anisotropy correlation decay [**61**], perturbed angular correlation gamma-spectroscopy [**62**], fluorescence lifetime [**63**]). The term "micro-viscosity" is also often used, however, this term refers to the viscosity experienced by micrometer-sized colloidal particles (experimentally measured for example by dynamic light scattering [**57**]). The term macro-viscosity refers to the viscosity measured with a rheometer. Not surprisingly, the distinction between these various definitions is washed out in quite a few publications. Our work is closely related to nano-viscosity as described by the fluorescence lifetime, where the influence of the viscous environment on the internal dihedral rotational dynamics of a fluorophore is measured. Clearly, the definition of nano-viscosity depends on the time over which the probe molecular motion is measured (be it translational or rotational) and, for an inhomogeneous medium, the position of the molecule or, for a liquid, the solvation environment around the probe molecule. Our work is about the relation between the solvent viscosity and the mean dihedral barrier-crossing time, which is rather complex since the mean dihedral barrier-crossing time is a measure of the solvent viscosity on time scales at which the dihedral barrier crossing takes place. In the previous version of our paper, we ruled out effects due to the inhomogeneous solvation environment around the probe molecule based on the fact that the friction coefficient γ for the dihedral angle extracted from simulations of butane in water-glycerol mixtures and in pure super-heavy water as a function of the macroscopic viscosity η (previous version Figure 2C, revised version Figure 2E) is the same. The rationale behind this is that super-heavy water, which is a single-component liquid, is homogeneous, and therefore, by the close agreement of the friction coefficients, the possible inhomogeneity of water-glycerol mixtures is small or irrelevant. However, this argument provides a sufficient but not a necessary condition, as is illustrated by the fact that for newly-introduced end-to-end reaction coordinate d_{14} , which we now additionally analyze in the revised version, the relation between the friction coefficient γ and the macroscopic viscosity η in Figure 2E differs for water-glycerol mixtures and pure super-heavy water. This does not mean that the solvation environment of butane in water-glycerol mixtures is inhomogeneous, but rather that the dependence of the friction on the viscoelastic solvent time scales differs.

In the revised version, we employ a much more direct argument to rule out the possibility that the inhomogeneous solvation environment around butane causes the breakdown of the Stokes-Einstein relation: For this, we consider the translational diffusion constant, derived from the long-time translational diffusivity of butane, as shown in the inset of Figure 2E (inset of Figure 2C in the previous version). The fact that the translational diffusion constant of butane in water-glycerol mixtures and pure super-heavy water agree quantitatively and both scale linearly in the macroscopic viscosity not only demonstrates that the Stokes-Einstein relation holds for translational motion, but also that the solvation environment around butane in water-glycerol mixtures is homogeneous and corresponds to bulk solvent conditions: If, for example, water would preferentially bind to butane, or if the solvating interfacial-water or glycerol molecular dynamics differed from bulk, the translational diffusion constant of butane would depend on the glycerol concentration and thus on the macroscopic viscosity in a non-linear fashion. In the revised version we moved mentioning of the terms nano-viscosity and micro-viscosity into the discussion section, where they are clearly defined as

outlined above, and argue that the solvation environment around butane in water-glycerol mixture is spatially homogeneous along the lines presented here.

3. Figure 2C is unconvincing - deducing scaling laws when viscosity changes by merely an order of magnitude is dubious. For example, in the inset the authors argue that they see direct proportionality to viscosity, but one could equally claim some other power law, as some curvature is observed. And while all plots tend to look like straight lines on the Log-log scale, the main Fig. 2C doesn't even quite look like a straight line!

We agree with the reviewer that it is misleading to deduce a scaling law from the given data. It was not our intention to deduce such a law, but rather simply to indicate that the relationship between the friction acting on a dihedral angle and the solvent viscosity is significantly sub-linear. The inclusion of an explicit scaling relation was rather intended to provide a visual guide for the approximate scaling exhibited by the data. We have removed the arguments for an explicit scaling law from this section since the main point can be conveyed without it. Furthermore, we have now replotted the data on a linear-linear scale, such that there is no attempt to pass the data off as a straight line on a log-log plot. We note, however, that we have maintained the linear scaling guide for the transitional diffusion, despite the slight deviation for high viscosities, since this is known to be the expected scaling in the absence of effective viscosity.

Reply to Reviewer #3

Relevant points from the reviewer's summary:

Previous work, in particular that of de Sancho and Best (2014) (ref. 12), which was also highlighted by the authors, has addressed the same questions and reached similar conclusions.

While we agree that there is overlap between our investigation and the previous work by de Sancho and Best, we suggest that our work is fundamentally different and that, in key instances, we actually reach different conclusions. The outstanding contribution from de Sancho et al. comes from their extensive simulation set of a variety of different protein models and peptides, and therefore from their ability to broadly explore the deviations from the inverse-linear relation $\tau \propto 1/\eta$ for many systems, which, like us, they consider for raw dihedral angles. While the subject matter of their paper is to discuss internal friction effects in proteins, defined as deviations from $\tau \propto 1/\eta$, the authors do not at any point calculate the friction acting on a protein. Defining internal friction via $\tau \propto 1/\eta$ is indirect since it neglects friction, which is the mutual quantity enabling this relationship via the Stoke-Einstein relations (a linear relationship between friction and viscosity) and a linear relationship between friction and reaction time (Kramers relation). Furthermore, except in a single instance, the authors only consider the super-heavy water method, which we show to be fundamentally problematic when interpreting the viscosity dependence of small-molecule reaction kinetics. In the single instance where the authors use a viscogenic agent (in their case glucose), we reach a different conclusion by showing that, in general, kinetic effects deviate between super-heavy water and mixed solvents. While it is true that the authors do at one point calculate memory friction, it is in a simplified context of a single Leonard-Jones particle moving on a 1D periodic free-energy surface. Their conclusion that the Grote-Hynes theory is an accurate model for this system is certainly an important result, but they can only qualitatively infer the applicability to dihedral isomerization. We explicitly extract the friction acting on a dihedral for a range of solutes and hence systematically quantify deviations from the Stoke-Einstein relation, Kramer's theory, and the Grote-Hynes theory (which was not possible for de Sancho and Best). In doing so, we reveal complex solvent-dependent effects, which are overlooked, or misinterpreted, in previous studies, but are essential for a full understanding of solute-solvent coupling. Furthermore, by having access to the dihedral memory friction, we can discuss the role of memory-induced kinetic acceleration, which is void from all previous discussions. None of these things are discussed by de Sancho and Best.

Overall, the conclusion that non-Markovian effects are partly responsible for internal friction effects in dihedral angle dynamics is not entirely new. This point has also been discussed for butane by one of the authors in a previous publication (ref. 14, Daldrop et al. PNAS, 2018), although I got the impression that the explanation of internal friction effects in that study was slightly different.

There are differences between our interpretation of internal friction in the present article compared to our previous work, which stems from the differences in scope between the two investigations. In our previous PNAS article, we only considered butane in super-heavy water, where position constraints were used to probe internal friction mechanisms. In the present article, we are explicitly considering the role of molecular-size effects with a focus on the role of solvent composition. In Fig.3E, we quantitatively show different characterizations of internal friction effects and that the combination of solute size and solvent compositions dictate which characterization defines a given system. In Fig.S3, we show that the two investigations are actually in agreement over the range of viscosity studied here ($\eta/\eta_0 \geq 1$), for super-heavy water. Our interpretation in terms of Eq.2 is consistent over a range of solute sizes, viscosities, and solvent compositions. Therefore, the conclusions reached in the present article were not possible in our previous work. We have endeavored to clarify these differences in the revised paper.

Reviewer comments:

1. This is a technical comment. I had serious problems understanding the handling of re-crossing (S7). I understand that barrier re-crossing effects need to be handled carefully when computing FP times. However, the authors seem to be concerned about re-crossing of states (in free energy minima) as suggested in their explanation in S7. The explanation of why this is a concern was unclear to me and should be better explained. In this context, the value of the delay δt and examples of FP time distributions for the other cases (All-to-First, Delayed All-to-First) should be given.

We appreciate the reviewer's comment and we have made steps to clarify these ambiguities. This is a subtle point and the subject of a forthcoming paper from our group. We are indeed concerned with the "re-crossing of states", as expressed by the reviewer, rather than "barrier re-crossing". The latter refers to instances when a system makes multiple crossings of a transition state in a single excursion between the reactant and product state, which is commonly discussed in the literature (in the context of protein folding, for example). We have checked carefully that this type of recrossing never happens for any of the dihedral rotations discussed in this paper. The "re-crossing of states", as is described in the SI section-S7, is particularly pronounced in our systems, especially for butane in the super-heavy water solvent. These recrossing processes manifest as fast, multi-exponential contributions to the passage time distributions. One can generate various distributions for the passage times. In the SI section S7, we distinguish between the all-to-first passage distributions, which are typically considered for mean FP time calculations, and first-to-first distributions, which are often referred to as waiting-time distributions. For butane and any system with significant contributions from "barrier re-crossing" effects, the mean values are dramatically affected depending on the choice of distributions. This is not the case for overdamped Markovian systems, which exhibit single-component exponential barrier-crossing-time distributions. We chose to present the mean delayed all-to-first passage times in the main article since it emphasizes the coupling of the longest exponential components of the barrier crossing process (effectively filtering out the contributions to the distributions from the rapid "re-crossing of states") to the viscosity, which couples quite strongly in the super-heavy water but not at all in the water-glycerol solvent. It is important to note that this filtering maintains the time scale of the longest exponential contribution, which is present in both the all-to-first distributions and the first-to-first. We have now largely rewritten the Supplementary Information section S7 and have updated Fig. S6 to present the various distributions, and to show the influence of the choice of distribution on the resulting mean barrier crossing time. We also explicitly state that $\delta t = 25$ ps for all simulations, which we seem to have lost during our draft editing processes. We thank the reviewer for pointing out this key piece of missing information.

2. Do the authors have an explanation for the different scaling of the MFP time with the number of backbone atoms for cis-trans and trans-cis transitions in alkanes? It seems that cis-trans transitions are less sensitive to the number of backbone atoms. Yet, the potentials for the individual alkanes are very similar.

The different scaling that the reviewer has indicated (Fig. 3D of the main manuscript) is for the inner dihedral of the alkane chains. While the reviewer is correct that the cis-trans transition (now referred to as the gauche-trans transition) appears less sensitive in the absolute sense, we find that this is not the case when we consider the relative sensitivities, which we suggest are more insightful. In the additional figure Fig.R2A, included as a part of this response letter, we see that the relative sensitivities are actually quite similar. We re-plot the data given in Fig. 3D with each τ_0 rescaled but the butane value $\tau_{0, \text{But}}$. Here, we see that, overall, the gauche-trans transition is only slightly less sensitive to the number of backbone atoms. If, however, we disregard butane (n -backbone = 4), which is a specific case since the butane molecule cannot exhibit multi-

dihedral effects, then the gauche-trans transition is actually slightly more sensitive than the trans-gauche (n-backbone = 6,8, and 10). These results are consistent with the relative changes observed in the free-energy barrier heights as a function of the number of backbone atoms (Fig.R2B). Here, the gauche-trans barrier decreases overall while the trans-gauche barrier increases overall. Considering just n-backbone = 6,8, and 10 then we see the opposite effects that the gauche-trans barrier slightly decreases and the trans-gauche barrier increases. Therefore, when we consider the relative barrier height changes and the relative sensitivity of transition times, the trends are in agreement. We have included this additional figure in the Supplementary Information Fig.S8 to clarify this point.

Fig.R2: (A) Barrier crossing times in neat water τ_0 , plotted as a function of n-alkane backbone length (in numbers of carbon atoms), for the inner-most dihedral of each n-alkane chain. To observe the relative sensitivity, we rescale by the reaction times for butane $\tau_{0, \text{But}}$. Therefore, we can observe the relative sensitivity for the two different transitions (gauche-trans and trans-gauche) as a function of backbone length. (B) Relative changes in free-energy barrier heights as a function of n-alkane backbone length. ΔU is the barrier height in either state and ΔU_{But} is the corresponding height for butane. We calculate the Boltzmann factor associated with each transition and rescale by the butane value.

3. I find the analysis of the timescales in the friction kernel very interesting (Fig. 4B). However, the discussion accompanying this figure is rather qualitative. It should be possible to omit some of the decay components in the kernel and then use Langevin simulations to generate trajectories and from those MFP times to see which decay component really dominates the viscosity dependence. This would be particularly interesting for butane.

This is an excellent point and we have now implemented precisely the reviewer's suggestion for the butane dihedral angle. In a newly introduced plot (Fig.4C) we show results for a GLE simulation, implemented using Markovian embedding techniques, for a 1D particle moving on the Ryckaert-Belleman potential. We include a memory kernel comprised of three decaying exponential terms, which are parametrized by the fitting results for butane in pure water (we neglect the oscillating component). We show that by systematically perturbing the amplitudes and time scales of the individual components of the memory kernel, the barrier crossing times are only sensitive to changes in a single memory component. This is the slowest component for which the time scale is less than the diffusion time. The results are in agreement with the arguments that we make in this section for the butane dihedral in MD simulations.

4. Related to point 3, the friction kernel contains information about the solvent. Have the authors thought about directly studying the solvent dynamics around, let's say, butane compared to bulk solvent? This would be helpful to understand what actual molecular processes in the solvent cause the individual components in the friction kernel.

We thank the reviewer for this helpful suggestion and we have now included an additional figure and Supplementary Information S13, where we do precisely this. As a measure for the solvent dynamics, we have calculated the scale times for water molecules to transition from the first hydration shell to the second shell for bulk water-water interactions. We also calculate the hydration-shell transition times for water molecules that are solvating the butane molecule. We show that the solvent dynamics are certainly modified around the butane molecule compared to their bulk values. Interestingly, the solvent relaxation time scales measured in this way are greater than the longest time scales detected in the memory kernels. This is especially true for the escape times for water from the hydration shell around the CH₂ groups, which can be greater than 50 ps for the high-viscosity solvents. Likewise, we now also show the longest time scale for the relaxation of the stress autocorrelation functions used to calculate the viscosity (Supplementary Information section S2), which are of the same order as the hydration-shell relaxation time scales.

Reviewer #1 (Remarks to the Author):

I thank the authors for thoroughly addressing my comments, and for their response to the comments of the other reviewers. I am pleased with all of the responses save one. The authors state:

"The authors calculate the transition rate using the population decay/reactive flux method of Chandler. This method has not been validated for non-Markovian systems. In Fig.R1 of this response letter, we show that the standard Chandler method of identifying a long relaxation timescale in the population decay correlation breaks down at moderate memory times, which is the regime for butane isomerization. We have now included a reference to Berne and coworkers (Rosenberg, Berne, and Chandler, Chem. Phys. Lett. 1980). However, we suggest that many important questions remain."

The reactive flux method is designed to provide a quantitative estimate of the transmission coefficient, a prefactor that corrects the transition state theory estimate of the reaction rate by accounting for recrossings of the transition state surface. As noted by Chandler (and by Yamamoto before) the method will be accurate as long as there is a significant separation in the time scales of the dynamics of barrier recrossing and the time scale for transition between reactant and product.

I believe this is completely general and is true whether the underlying dynamics is Markovian or non-Markovian. In fact, there are numerous examples of the accuracy of the method when applied to systems with strongly non-Markovian features (<https://doi.org/10.1063/1.450425>). I would appreciate hearing the thoughts of the authors.

Reviewer #2 (Remarks to the Author):

the authors addressed the comments from all the reviewers. I recommend publication of this manuscript

Reviewer #3 (Remarks to the Author):

I would like to thank the authors for addressing my points so rigorously. Particularly the analysis of the memory kernel impact is really neat and I appreciate the work. The paper is now ready for publication.

Response to Referee's comments: "The role of memory-dependent friction and solvent viscosity in isomerization kinetics in viscogenic media"

Reply to Reviewer #1

I thank the authors for thoroughly addressing my comments, and for their response to the comments of the other reviewers. I am pleased with all of the responses save one. The authors state:

"The authors calculate the transition rate using the population decay/reactive flux method of Chandler. This method has not been validated for non-Markovian systems. In Fig.R1 of this response letter, we show that the standard Chandler method of identifying a long relaxation times-scale in the population decay correlation breaks down at moderate memory times, which is the regime for butane isomerization. We have now included a reference to Berne and coworkers (Rosenberg, Berne, and Chandler, Chem. Phys. Lett. 1980). However, we suggest that many important questions remain."

The reactive flux method is designed to provide a quantitative estimate of the transmission coefficient, a prefactor that corrects the transition state theory estimate of the reaction rate by accounting for recrossings of the transition state surface. As noted by Chandler (and by Yamamoto before) the method will be accurate as long as there is a significant separation in the time scales of the dynamics of barrier recrossing and the time scale for transition between reactant and product.

I believe this is completely general and is true whether the underlying dynamics is Markovian or non-Markovian. In fact, there are numerous examples of the accuracy of the method when applied to systems with strongly non-Markovian features (Straub et al. J. Chem. Phys. 1986). I would appreciate hearing the thoughts of the authors.

We would like to thank the reviewer for their extensive critical feedback overall and we are very glad to learn that they are satisfied with our revised manuscript. The reviewer is right to be so persistent on this final point and we can assure both the reviewer and the editor that we have spent much time and energy thinking about these things. The reactive-flux method is indeed designed to provide a quantitative estimate of the transmission coefficient, as the reviewer points out. In its classical form, it provides a numerical recipe to extract an estimate for a barrier crossing rate, given by

$$k_{r \rightarrow p}(t) = \frac{1}{x_r} \langle \delta(x(0)) \dot{x}(0) H_p(x(t)) \rangle \quad \text{Eq.1}$$

which is implemented by initiating many instances of a particle or molecule from a transition state, where we have assumed that the transition state is located at $x=0$. x_r is the mean reactant-state occupation, $H_p(x)$ is the unit step function that is 1 in the product state and 0 in the reactant state, and $\langle \dots \rangle$ represents the ensemble average taken over many realizations of the simulation. Chandler went to great lengths to show that under certain conditions the time-dependent rate predicted by the reactive-flux formalism is equivalent to the phenomenological rate equation due to first-order reaction kinetics, a relationship which he founded based on Onsager's regression hypothesis. Since, in most realistic situations, the direct equivalence is not true, Chandler was satisfied to say that the relationship holds on certain time scales, i.e. times scales that are greater than time scales of the dynamics of barrier recrossing and less than the time scale for transition between reactant and product, as stated by the reviewer. Thus, Eq.1 is only applicable when the argument t falls within a specific range. Assuming this separation of time scales, the reactive-flux formalism is applicable in many dynamic regimes (e.g. overdamped, inertial, or, presumably, memory-dependent regimes).

We previously made the statement that "this method has not been validated for non-Markovian systems". We concede that this statement is perhaps too broad. Therefore, we would like to take this opportunity to specify our intentions. It is certainly true that the reactive-flux method has been applied to non-Markovian systems and hence the rates determined by the method have been presented for a range of memory-relaxation time scales, and over a range of separation-barrier heights and damping strengths. The reviewer has indicated the classical work of Straub, Borkovec, and Berne (J. Chem. Phys. 1986), where a 1D Brownian simulation in a double-well potential with an explicit non-Markovian friction kernel was used to validate a particular theoretical prediction for barrier-crossing reaction kinetics which should, in principle, account for memory-

dependent friction effects. The authors applied the computationally efficient form of the reactive-flux formalism, namely the rapid absorbing boundary method, which they show to be equivalent to the classical version (Eq. 1), and hence confirmed that current theories, which combine the Grote-Hynes rate predictions with predictions for energy-diffusion dominated processes, are not generally valid for describing the measured rates. However, the authors do not provide a complete theory to describe the measured reaction rates, as they are evaluated from the reactive-flux formalism. Furthermore, they do not compare the rates obtained from their reactive-flux calculation to other numerical results, such as might be obtained from a mean first-passage time (MFPT) calculation or an explicit escape time calculation, which can also be performed using model simulations with an absorbing boundary condition. The authors merely suggest that one way forward would be to pursue a framework that accounts for the effective dynamic potential, which they introduce but do not pursue. Therefore, it is hard to say that the authors “validated” the reactive flux formalism. Rather, they were satisfied to present the numerical results and to use these results to invalidate known theories, while stopping short of connecting their results to other measures of a transition time.

In the intermediate-to-long memory time regime, which is explored by Straub et al., it is known that, when measured using the MFPT, non-Markovian systems exhibit memory-induced reaction slow-down effects (Kappler et al. *J. Chem. Phys.* 2018). The Grote-Hynes theory, for example, does not predict this slow-down effect since it only considers particle dynamics at the transition state, rather avoiding the time spent in a well as a particle awaits activation, which is accounted for by MFPTs and escape times. This memory-induced slowdown regime is relevant for the folding and unfolding dynamics of some small proteins (Dalton et al. *PNAS*, 120 (31) e2220068120, 2023). To our understanding, it has not been investigated whether the reactive-flux formalism is suitable for predicting such known non-Markovian reaction slow-down effects. We presume that the method is not suitable for this task since, much like the Grote-Hynes theory, the method is applied to ensembles of particles that are initiated at the transition state.

In our previous reply to the reviewers, we considered the disclaimer that the reactive-flux formalism is only suitable when there is a clear separation of time scales between recrossing processes and the transition between reactant and product. To our understanding, the validity of this condition is satisfied when a distinct plateau region can be identified in the rate function $k(t) = dC(t)/dt$, where $C(t)$ is evaluated using the time correlation for the population operator (described in detail in many treatments of this topic). Using the same 1D model employed by Straub et al., we showed that this separation of time scales (i.e., the identification of a distinct plateau region in $k(t)$) is not possible once memory relaxation times are significant. We have included the figure from our previous reply here again for completeness (Fig.R1).

For Markovian systems, the relation between MFPT and escape time is $\tau_{\text{esc}} = 2\tau_{\text{MFP}}$ (Hänggi, Talkner, Borkovec 1990), and it is assumed that the total relaxation rate $k_{\text{rx}}^{-1} = (k^+ + k^-)^{-1}$, where k^+ and k^- are the associated forward and backward rates evaluated by the reactive flux formalism, are related, such that for symmetric potentials, $k_{\text{rx}} = \tau_{\text{esc}}^{-1}$. Hänggi et al. themselves state that “... the plateau value, or its inverse, the plateau time, are not appropriate quantities to be calculated on analytical grounds. Thus, almost all of the analytical work discussed [in their treatment] rest upon the [... various methods including MFPT]”. They go on to state that “unfortunately, the MFPT is a rather complex notion for a general (non-Markovian) stochastic process”. Therefore, the relationship between all of these quantities in non-Markovian systems eluded Hänggi et al. at least in 1990.

Therefore, to conclude, when we stated that “this method has not been validated for non-Markovian systems”, we meant that for significantly non-Markovian systems, or systems where the memory time of the order of, or significantly longer than, the diffusion time τ_{D} , (defined as $\tau_{\text{D}} = L^2\gamma/k_{\text{B}}T$, where L is a characteristic length and γ is the total friction), it is not clear how the rates associated with the reactive flux formalism should related to the MFPT or escape time (and other definitions of transition times - see our recent arXiv paper for some comparisons Zhou et al., arXiv:2403.06604, 2024), which will likely return different values (as does the Grote-Hynes theory - see Kappler et al. 2018), since the MFPT is dominated by the segments of the trajectory spent in well. We believe that this knowledge gap is particularly relevant for complex molecular dynamics simulations, such as those presented in our submitted article.

We have added a reference to Straub, Borkovec, and Berne (*J. Chem. Phys.* 1986) in the revised article, with a brief comment regarding its relevance to our investigation.

Fig.R1: In the reactive flux method of Chandler, the decay rate $k(t)$, calculated via the population correlations $C_{HH}(t)$, should reveal a distinct plateau regime, which is associated with the phenomenological reaction rate. Plotted in terms of $\tau(t) = k^{-1}(t)$, where $k(t) = -d(C_{HH}(t)/C_{HH}(0))/dt$, the plateau will appear as a region of constant τ . Here, using 1D GLE simulations performed with the Markovian-embedding technique (Kappler et al. JCP, 2018), we compare $\tau(t)$ to the reaction times evaluated via the mean first-passage times (see Supplementary Information section S7). We show that for moderate memory times ($\tau_r/\tau_D > 0.01$, where τ_D is the diffusion time scale), it is impossible to identify a distinct plateau region in $\tau(t) = k^{-1}(t)$. Even for small memory times ($\tau_r = 0.001\tau_D$), the plateau region is very short. The two mean first-passage times are the first-to-first (τ_{fp}), which only uses information about first arrivals into states and is equivalent to the waiting time, or dwell time, and the all-to-first (τ_{allfp}), which uses information about every crossing of a state minima before traversing a barrier. Up until $\tau_r = 0.033\tau_D$, the small plateau region coincides with both mean first-passage time calculations. However, for longer memory times, where the two mean first-passage time methods are known to diverge (Zhou et al. arXiv:2403.06604, 2024), there is no agreement at all. Therefore, the applicability of Chandler's reactive flux method involving population-operator decay to non-Markovian systems is unclear.